# Merkel Cell Polyomavirus in the Context of Oral Squamous Cell Carcinoma and Oral Potentially Malignant Disorders

**DOI:** 10.3390/biomedicines12040709

**Published:** 2024-03-22

**Authors:** Sara Passerini, Giulia Babini, Elisabetta Merenda, Raffaella Carletti, Daniela Scribano, Luigi Rosa, Antonietta Lucia Conte, Ugo Moens, Livia Ottolenghi, Umberto Romeo, Maria Pia Conte, Cira Rosaria Tiziana Di Gioia, Valeria Pietropaolo

**Affiliations:** 1Department of Public Health and Infectious Diseases, “Sapienza” University of Rome, 00185 Rome, Italy; babini.1398682@studenti.uniroma1.it (G.B.); daniela.scribano@uniroma1.it (D.S.); antoniettalucia.conte@uniroma1.it (A.L.C.); mariapia.conte@uniroma1.it (M.P.C.); 2Department of Radiological, Oncological and Pathological Science, “Sapienza” University of Rome, 00161 Rome, Italy; elisabetta.merenda@uniroma1.it (E.M.); raffaella.carletti@uniroma1.it (R.C.); cira.digioia@uniroma1.it (C.R.T.D.G.); 3Laboratory of Virology, National Institute for Infectious Diseases “Spallanzani”, 00149 Rome, Italy; luigi.rosa@inmi.it; 4Department of Medical Biology, Faculty of Health Sciences, University of Tromsø, UiT-The Arctic University of Norway, 9037 Tromsø, Norway; umo000@uit.no; 5Department of Oral and Maxillofacial Sciences, Sapienza University of Rome, Via Caserta 6, 00161 Rome, Italy; livia.ottolenghi@uniroma1.it (L.O.); umberto.romeo@uniroma1.it (U.R.)

**Keywords:** Merkel Cell Polyomavirus, oral cancer, oral diseases, miRNAs, LTAg, VP1

## Abstract

Despite recent advances in prevention, detection and treatment, oral squamous cell carcinoma (OSCC) remains a global health concern, strongly associated with environmental and lifestyle risk factors and infection with oncogenic viruses. Merkel Cell Polyomavirus (MCPyV), well known to be the causative agent of Merkel Cell Carcinoma (MCC) has been found in OSCC, suggesting its potential role as a co-factor in the development of oral cavity cancers. To improve our understanding about MCPyV in oral cavities, the detection and analysis of MCPyV DNA, transcripts and miRNA were performed on OSCCs and oral potentially malignant disorders (OPMDs). In addition, the cellular miR-375, known to be deregulated in tumors, was examined. MCPyV DNA was found in 3 out of 11 OSCC and 4 out of 12 OPMD samples, with a viral mean value of 1.49 × 10^2^ copies/mL. Viral integration was not observed and LTAg and VP1 transcripts were detected. Viral miRNAs were not detected whereas the cellular miR-375 was found over expressed in all MCPyV positive oral specimens. Our results reported evidence of MCPyV replication in both OSCC and OPMD suggesting the oral cavity as a site of replicative MCPyV infection, therefore underscoring an active role of this virus in the occurrence of oral lesions.

## 1. Introduction

Oral cancer (OC) is the most common malignant neoplasm in the head and neck (HN) region and the 16th most common cancer worldwide [1,2]. Among OCs, oral squamous cell carcinoma (OSCC) is the most prevalent and the mouth and tongue are the most frequent sites for the development of this type of tumor [3]. OSCC can arise de novo or from oral potentially malignant diseases (OPMDs) such as leukoplakia, proliferative verrucous leukoplakia (PVL), erythroplakia, actinic keratosis, oral lichen planus (OLP) and oral submucous fibrosis with different potentials for malignant transformation [4]. Oral oncogenesis is a complex process that involves several risk factors such as smoking, alcohol consumption, nutrition, and genetic factors [5]. Moreover, viruses have been proposed as a causal factor because oncoviruses such as High-Risk Human Papillomaviruses (HR-HPV) and Epstein–Barr Virus (EBV) have been detected in oral cavity tumor tissues [3,6].

Merkel Cell Polyomavirus (MCPyV) is a human polyomavirus which is a natural part of the skin microbiome, but viral DNA has also been isolated from respiratory, urine and peripheral blood samples [7,8,9]. Serological studies have shown that MCPyV is common in humans with seroprevalences in around 80% of healthy adults [10,11]. Primary infection usually occurs in childhood and persists asymptomatically throughout adult life [12]. MCPyV transmission route remains to be clarified and could involve mucosal, cutaneous, fecal–oral or respiratory ways [12].

The MCPyV circular double stranded DNA genome comprises the early and late gene region on opposite strands and the Non-Coding Control Region (NCCR) located between them [12]. The early region encodes the non-structural proteins Large T (LT) and small T (sT) antigens, which are involved in regulation of viral replication and transcription, and the 57 kT antigen and for ALTO proteins whose functions are still unclear [13]. The late region encodes for the capsid proteins viral protein 1 (VP1) and viral protein 2 (VP2) and for mature microRNAs (miRNAs), referred to as mcv-miR-M1-5p and mcv-miR-M1-3p which can modulate early gene expression and thus regulate the viral cycle and promote the host immune evasion [12,14,15]. Interposed between the early and late regions there is the NCCR, that contains the origin of replication (Ori) and the promoters and enhancers for regulation of early and late genes expression. The NCCR is a hypervariable region whose polymorphism has been described [16]; however, whether MCPyV NCCR variations have an influence on viral replication and pathogenic properties remains to be examined [13].

Although MCPyV infection is usually asymptomatic, studies in the past decade have established MCPyV as the major causative factor of Merkel Cell Carcinoma (MCC), an aggressive neuroendocrine carcinoma of the skin, occurring in about three people per million members of the population [7,17]. Initially described as trabecular carcinoma of the skin, it is also known as cutaneous APUDoma, primary neuroendocrine carcinoma of the skin, and primary small cell carcinoma of the skin [17]. Approximately 80% of all examined MCC samples are positive for MCPyV. Specifically, in MCPyV-positive tumors, viral genome integration occurs and a C-terminal truncated form of LTAg is observed, whereas a wild-type sTAg is expressed [18,19]. Truncation retains LTAg’s ability to interact with with the Retinoblastoma protein (pRb) but ablates the binding with p53 [20]. Studies with transgenic mice demonstrated that the truncated LTag-pRb interaction is important for tumorigenesis [21]. In contrast to TAgs from other polyomaviruses, where LTAg is the most dominant transforming oncoprotein and sTAg exert an auxiliary function, in vitro studies showed that MCPyV sTAg but not LTAg, is able to induce transformation in rodent fibroblasts and to sustain proliferative signaling by activating MYCL Proto-Oncogene, basic Helix-Loop-Helix (bHLH) transcription factor [22,23,24]. Moreover, sTag is oncogenic in transgenic mice models [23,25,26], suggesting that sTAg is the main oncoprotein [13]. The biological relevance of the MCPyV-encoded miRNAs in tumorigenesis is still unclear because mcv-miR-M1-5p expression in viral positive MCCs is low, absent, or non-detectable [15].

Considering the oncogenic potentials of MCPyV and its widespread prevalence among the human population, several studies have been carried out to elucidate a plausible role of MCPyV in the development of non-MCC tumors. MCPyV has been identified in various other cancers, amongst them head and neck tumors (HN), including esophagus and oral cavities [7,27,28,29,30,31,32]. However, in most cases the presence of viral DNA was examined, whereas expression of viral genes, state of the viral genome, and the possible truncation of LTAg was not investigated. Moreover, several studies reported MCC in the oral region, suggesting including this kind of tumor in the differential diagnosis of lesions arising in the mucosa of the HN [33,34,35].

Because the incidence and possible role of MCPyV in oral cancer remains elusive, the purpose of our study was to investigate the presence of the viral DNA and RNA and to examine integration and truncation of LT, two hallmarks of MCPyV-positive MCCs, in oral squamous cell carcinoma (OSCC) and oral potentially malignant disorders (OPMD). Although MCPyV DNA was present and viral genes were expressed in some samples of both clinical conditions, the absence of integrated MCPyV and truncated LT may jeopardize a role in oral cancer or suggest a different modus operandi in MCPyV-induced oral cancer. In addition, the viral miRNAs’ expression was investigated, in order to verify their potential role as biomarkers for MCPyV infection, as well as the cellular miR-375, well known to be deregulated in tumors.

## 2. Materials and Methods

### 2.1. Clinical Specimens

Formalin-fixed paraffin-embedded (FFPE) tissues retrieved from the Department of Pathology of Policlinico Umberto I—Sapienza University of Rome (Rome), were obtained from 11 OSCC and 12 OPMD patients (9 males and 14 females, age range 26–82 years, mean age 59.2 ± 15.7) (Table 1). Histopathological diagnosis was performed by a pathologist dedicated to oral pathology. For the histological diagnosis, five groups of oral pathology were defined: (1) OSCC, (2) keratosis with dysplasia, (3) keratosis without dysplasia, (4) PVL and (5) OLP (Table 1).

This study was conducted retrospectively from data obtained for clinical purposes. We consulted extensively with the Ethic Committee Sapienza University of Rome, Policlinico Umberto I who determined that our study did not need ethical approval.

### 2.2. Histological Diagnosis

According to the WHO Classification of Head and Neck Tumors 2022 (beta version), histologic diagnosis of OSCC was made when the oral lesion showed a malignant neoplasia that arises from the mucosal stratified squamous epithelial that shows dysplasia and extends through the basement membrane and into the underlying fibrous connective tissue, with variable squamous differentiation grading as well (G1), moderately (G2), and poorly differentiated (G3), based on the amount of keratinization, mitotic activity, cellular and nuclear pleomorphism and pattern of invasion (Figure 1A). For histologic evaluation of OPMDs, the absence or presence and the degree (mild, moderate, severe) of epithelial dysplasia characterized by cytologic and architectural atypia were evaluated in all the keratotic lesions reported clinically as leukoplakia/erytroplakia (Figure 1B). The diagnosis of PVL was made in some oral epithelial lesions characterized by a wide spectrum of histological changes indicated in three categories by the Expert Consensus Guideline of Thompson and colleagues [36]: “corrugated ortho(para)hyperkeratotic lesion, not reactive”; “bulky hyperkeratotic proliferation, not reactive”; and “suspicious for” or “squamous cell carcinoma”. The integration of clinical–pathologic data is required for the PVL diagnosis (Figure 1C). Histologically, OLP lesions showed ortho(para)hyperkeratosis, apoptotic keratinocytes, interface mucositis with a band-like zone of mainly lymphocytic infiltration confined to the superficial lamina propria, and liquefactive degeneration in the basal cell layer. The diagnosis of OLP is based on both clinical and histopathological features.

### 2.3. DNA Extraction

Following deparaffinization with xylene, total DNA was extracted from FFPE biopsies using Quick-DNA FFPE kit (Zymo Research, Irvine, CA, USA), according to the manufacturer’s instructions. The extracted nucleic acid was eluted in a final volume of 50 µL and then evaluated for its PCR suitability by amplifying the *β-globin* gene sequences [37].

### 2.4. Detection of MCPyV DNA by Real-Time PCR

Detection and quantitation of MCPyV DNA in FFPE sections were performed by a quantitative polymerase chain reaction (qPCR) employing the previously described primers and probe for the MCPyV *sT* gene (forward: TTAGCTGTAAGTTGTCTCGCC; reverse: CACCAGTCAAAACTTTCCCAAG) [38]. All qPCRs were carried out in triplicate, and MCPyV viral load, given as the mean of at least three positive reactions was expressed as copies/milliliter. The number of viral copies was calculated from standard curves, which were constructed using a ten-fold dilution series of plasmid pMCV-R17a containing the entire genome of MCPyV (Addgene, #24729) (dilution range: 10^8^–10 copies/mL). Each qPCR session included negative and positive controls.

### 2.5. Standard PCR for MCPyV LTAg, NCCR and VP1 and Sequencing

MCPyV DNA positive samples were then subjected to standard polymerase chain reaction (PCR) with different sets of primers targeting MCPyV *LTAg* (LT1 and LT3), NCCR and *VP1* regions [39,40]. The PCR products underwent electrophoresis on a 2% agarose gel and were visualized using SafeView reagent (Applied Biological Materials, Vancouver, BC, Canada) under UV light (see Appendix A). In addition, in order to examine NCCR and *VP1* sequence variations, the amplified products were purified using an miPCR purification kit (Metabion, Plannegg, Germany) and sequenced in a dedicated facility (Bio-Fab research, Rome, Italy). The obtained sequences were compared to their reference strains deposited in GenBank (MCC350: EU375803). Sequence alignment was performed using Clustal W2 on the European Molecular Biology Laboratory–European Bioinformatics Institute (EMBL–EBI) website using default parameters [41].

### 2.6. Analysis of the MCPyV Integration Sites

Viral integration sites were evaluated by the detection of the integrated papilloma sequence (DIPS)–PCR technique which allowed for the amplification of the junctions between viral and cellular genomes [42]. Following DNA digestion with the TaqI restriction enzyme, the obtained DNA fragments were ligated to enzyme-specific adaptors and subjected to a PCR amplification employing viral- and adaptor-specific primers. The PCR products were then purified and sequenced. The integration sites were defined by submitting sequences to the databases of the National Center for Biotechnology Information and analyzing them with the Basic Local Alignment Search Tool (BLAST) for genomic localization [39,43].

### 2.7. Sequencing Analysis of MCPyV LTAg Gene

The DNA sequences from nucleotide positions 151 to 3102 (GenBank strain EU375803), corresponding to the entire *LTAg* gene, were determined by a PCR using a combination of six primer sets [39]. A direct sequence analysis was then carried out on the amplified products.

### 2.8. Analysis of MCPyV LT and VP1 Gene Expression

Total RNA, including miRNA, was extracted using the FFPE RNA purification kit (Norgen, Thorold, ON, Canada). Following RNA quality and quantity assessment by A230/A260 ratios, reverse transcription (RT) was performed by SensiFAST cDNA Synthesis kit (Meridian Bioscience, Cincinnati, OH, USA). An aliquot of the RT reaction mixture (1 μL) was used for the subsequent PCR amplification carried out with primer sequences to determine MCPyV *LT* and *VP1* gene expression [39]. Moreover, cDNA’s quality was confirmed amplifying the *β-globin* sequences.

### 2.9. Real-Time PCR for miRNA

Applied Biosystems™ TaqMan™ MicroRNA Assays (Thermo Fisher Scientific, Waltham, MA, USA) were used to measure relative levels of miRNAs expression according to the manufacturer’s instructions. Pre-designed TaqMan microRNA assays for mcv-miR-M1-5p (ID 006356) and miR-375 (ID000564) were used. Moreover, RNU6B (ID001093) was used as an endogenous control for normalization of miRNA expression. All reactions were performed in triplicate and relative miRNAs expression levels were calculated by the comparative ΔCt method and reported as 2^−ΔΔCt^.

### 2.10. Statistical Analysis

MCPyV detection was summarized by counts and proportions. The continuous variables were expressed both as mean ± SD and as median and range. The Χ^2^ test was performed to evaluate differences in the viral detection, and the Mann–Whitney U-test for non-normally distributed continuous variables was applied to analyze differences between patients. A *p* value < 0.05 was considered statistically significant.

## 3. Results

### 3.1. Detection of MCPyV DNA qPCR and Standard PCR

MCPyV sT DNA was found in 7/23 (30.4%) oral cavity biopsies with a viral load mean value of 1.49 × 10^2^ ± 7.65 copies/mL. Specifically, MCPyV DNA was detected in 3/11 (27.3%) OSCC and 4/12 (33.3%) OPMD samples with a viral load mean value of 1.17 × 10^2^ ± 7.5 and 1.74 × 10^2^ ± 99.2 copies/mL, respectively. The mean MCPyV DNA load, according to histopathological diagnosis is represented in Figure 2. MCPyV positivity was not associated with cancerous or non-cancerous oral lesions, nor a statistically significant difference was observed in the MCPyV viral load between OSCC and OPMD samples (*p* > 0.05) (Figure 2b).

The seven samples tested positive by qPCR were further positive for LT1 and LT3 amplification as well as for MCPyV NCCR and VP1 (Table 2).

### 3.2. MCPyV NCCR and VP1 Analysis

The amplified MCPyV NCCR and VP1 DNA fragments were compared with the reference sequence of the prototype North American MCC350, strain EU375803. Alignments of the seven NCCRs revealed a canonical structure in all analyzed samples. The sequencing analysis of MCPyV VP1 reported some nucleotide changes with respect to the reference strain, but none of them resulted in an amino-acid change in the derived protein sequence (Table 3). Specifically, one OSCC sample contained a 4179 C to A transversion whereas one OPMD displayed a 4204 T to C transition (Table 3).

### 3.3. Integration of Viral Genome and DNA Sequencing Analysis of MCPyV LT Gene

DNA from seven MCPyV positive samples was used for the analysis of viral integration and for the full-length *LT* gene (nucleotide positions 151–3102) sequencing. None of the samples reported MCPyV integration whereas a full-length *LT* gene sequence was amplified in all examined samples (Table 3). Moreover, a sequence analysis of the full-length *LT* gene did not reveal any mutations.

### 3.4. Expression of Transcripts from MCPyV LT and VP1 Genes

MCPyV positive biopsies were further subjected to transcripts analysis of MCPyV *LT* (nucleotide positions 910–1152) and *VP1* genes (nucleotide positions 3786–4137) by RT-PCR. All samples showed the expression of both *LT* and *VP1* genes (Table 2).

### 3.5. miRNA Expression

Oral biopsies were also tested for mcv-miR-M1-5p but none of the samples were found positive for MCPyV-encoded miRNA. Investigating the cellular miR-375, instead, a significant over-expression was observed in MCPyV-positive compared with negative samples (*p* < 0.05). Moreover, among virus positive samples, no evidence for a significant correlation was reported with neoplastic or non-neoplastic tissues (*p* > 0.05) whereas among the negative ones, miR-375 expression was significantly reduced in OSCC compared with OPMD samples (*p* < 0.05) (Figure 3).

## 4. Discussion

To date, no attributable etiologic factor has been proven for OSCC development. Therefore, investigating potential risk factors is highly important for OC diagnosis and prognosis. MCPyV has been clearly associated with MCC but viral DNA and transcripts have been also identified in other neoplasms, suggesting a potential role of this virus as a cancerogenic factor for non-MCC malignancies [7]. Specifically, fragments of MCPyV genome have been isolated in HN malignant tissue [29,32,44]. To investigate whether this oncogenic virus could be involved in the etiology of oral cavity tumors, in the current study, MCPyV prevalence and the molecular state were examined. MCPyV DNA was detected in both neoplastic and non-neoplastic oral tissue, suggesting that MCPyV positivity is not correlated with cancerous or non-cancerous oral lesions, as previously reported [30,45,46]. Moreover, the difference between viral load in OSCC and OPMD was not statistically significant; indeed, both tumor and non-tumor samples reported a low copy number of MCPyV DNA, about 1.49 × 10^2^ copies/mL, suggesting that the virus could establish a persistent replication in the oral cavity, without any pathological outcome [30].

In addition, sequence analysis was carried out on virus positive biopsies, in order to investigate NCCR variability in oral tissues. Since it is well known that NCCR rearrangements could increase viral replication and influence gene expression and virulence properties, the absence of sequence mutations could explain the low viral load detected in our MCPyV-positive samples [47]. Moreover, the isolation of canonical NCCRs even in tumor tissues suggests that NCCR rearrangements are not involved in carcinogenesis, confirming what was reported for MCPyV-positive MCCs [13]. Sequencing analysis was performed also on VP1 amplicons in order to evaluate possible amino acid changes that could influence MCPyV infectivity. A high degree of homology with the reference strain was observed also for VP1 sequences, whose nucleotide differences did not produce any amino acid change in the derived protein sequence.

However, since the detection and quantitation of viral DNA as well as the characterization of NCCR and VP1 regions are not sufficient to define the MCPyV role in oral malignancies, further analyses have been performed. In spite of what has been reported for virus-positive MCCs, where viral replication is hampered, late genes are not expressed [39,48], and transcript analysis revealed both *LT* and *VP1* gene expression in all virus-positive samples, supporting viral replication activity in our samples.

To date, several studies reported viral integration in the host genome and the expression of a truncated *LT* gene with a preserved LXCXE motif, able to bind and sequestrate pRb as key players in MCPyV-mediated oncogenesis [24,39,48]. In our MCPyV-positive samples, viral integration was not reported and a full-length *LT*, retaining both pRb and p53 binding domains, was observed. Besides LT truncation and viral integration, studies in vitro and in animal models demonstrated that MCPyV sT seems to be implicated in tumorigenesis, due to its capability to activate several oncogenic signaling pathways [49].

As sT’s transforming capacity is independent of LT expression [50], in oral cavity tissues in which qPCR reported sT detection, it is possible to hypothesize that it could be involved or contribute to stimulating tumor progression. However, further studies are required to improve our knowledge about sT’s role in OC.

Since it is well known that miRNAs may be relevant to understanding viral persistence, serving as a potentially comprehensive marker of infection status, in the current study the expression of viral miRNA mcv-miR-M1-5p has been investigated. Previous studies examining MCPyV-encoded miRNA in saliva, demonstrated a higher positivity of viral miR-M1-5p in the presence of a lower DNA status, suggesting a miRNAs’ potential role in regulating viral replication in oral cavity [51]. Moreover, since the modulation of miRNA expression may play a role in viral-associated diseases, MCPyV encoded miRNA has been examined also in the context of MCC, where a low expression has been reported [14] and this finding was further confirmed by in vitro studies on virus-positive MCC cell lines [52]. However, to date, the biological significance of viral miRNAs in tumorigenesis is still unclear. Analyzing oral biopsies for mcv-miR-M1-5p, all samples were found negative; therefore, its value as a biomarker and its relevance in OC remain doubtful.

In addition, since it is well known that the miRNA expression profile is frequently deregulated in pro-inflammatory environments and various cancer types, the expression pattern of the cellular miR-375 in virus-positive and negative samples was evaluated. MiR-375 is a tumor suppressor miRNA inhibiting cancerous properties such as proliferation, migration, and invasion in OC [53]. Indeed, previous studies demonstrated that a majority of oral tumors and HN cancer cell lines expressed lower miR-375 levels than any normal controls [53,54]. This miRNA, instead, is highly expressed in MCC; specifically, miR-375 levels were found to be significantly higher in virus positive than the virus-negative MCC tumors, suggesting its potential use as a surrogate marker for tumor burden in MCC [55]. Here we reported that miR-375 is highly expressed in MCPyV-positive biopsies compared with virus-negative samples with no statistically significant difference among cancerous and non-cancerous samples, suggesting a correlation between virus positivity and cellular miR-375 expression profile. Examining virus-negative samples, instead, OSCC revealed a significant lower expression of miR-375 compared with OPMD samples, confirming the importance of this miRNA as a tumor suppressor in OC [53].

## 5. Conclusions

In conclusion, our findings showed evidence of MCPyV replication in both OSCC and OPMD specimens, supporting the oral cavity as a site of replicative MCPyV infection. Although the low replicative form of MCPyV in this anatomical site weakens the hypothesis of a pathogenic role of the virus in oral malignancies, the persistent presence of MCPyV in oral tissues could lead to a possible active role of the virus in the occurrence of oral lesions. On the basis of these preliminary results, further studies including a larger cohort of patients are warranted in order to elucidate the pathogenic relevance of MCPyV in oral cavity lesions.

## Figures and Tables

**Figure 1 biomedicines-12-00709-f001:**
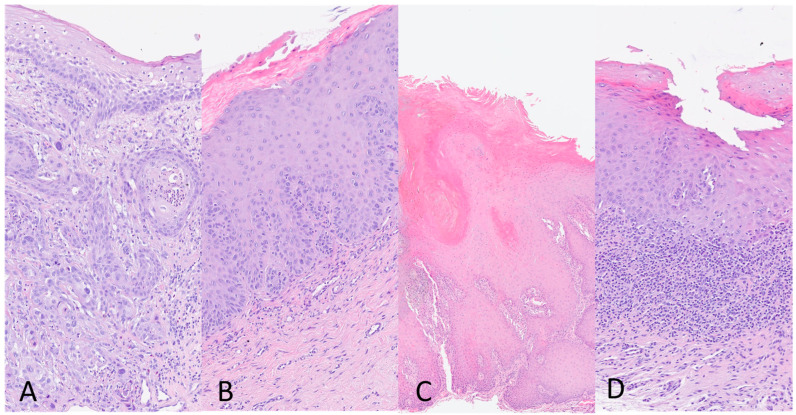
(**A**) Oral Squamous Cell Carcinoma (OSCC), moderately differentiated (G2) (10×); (**B**) Oral Keratotic lesion with Dysplasia (10×); (**C**) Proliferative Verrucous Leukoplakia (PVL) (10×); and (**D**) Oral Lichen Planus (OLP) (4×). Haematoxylin and eosin stain.

**Figure 2 biomedicines-12-00709-f002:**
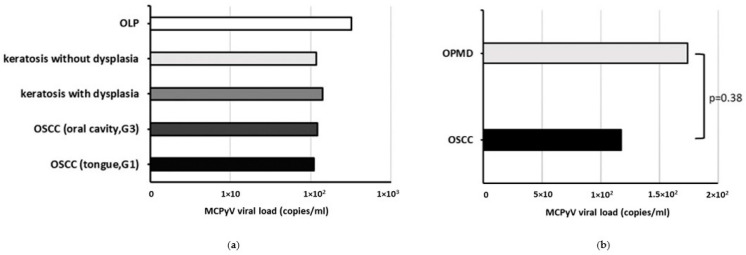
Mean MCPyV viral load according to histopathological diagnosis in patients with OSCC and OPMDs. (**a**) by type of lesion; (**b**) by type of samples.

**Figure 3 biomedicines-12-00709-f003:**
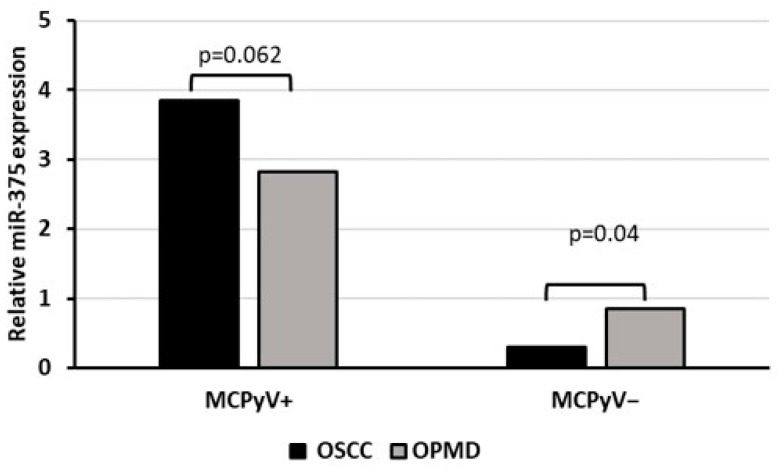
Relative miR-375 expression associated with MCPyV status in OSCC and OPMD samples.

**Table 1 biomedicines-12-00709-t001:** Demographic and clinical characteristics.

Features	Population	
Patients	23	
Mean age, years (SD)	59.2 (±15.7)	
Median age, years	62	
Gender, n (%)	M	F
	9 (39%)	14 (61%)
Histological diagnosis n (%)		
OSCC	11 (48%)	
Tongue	5 (22%)	
G1	1 (4%)	
G2	3 (13%)	
G3	1 (4%)	
Oral cavity	6 (26%)	
G1	2 (8.5%)	
G2	2 (8.5%)	
G3	2 (8.5%)	
OPMDs	12 (52%)	
keratosis with dysplasia	1 (4%)	
keratosis without dysplasia	3 (13%)	
PVL	4 (17%)	
OLP	4 (17%)	

**Table 2 biomedicines-12-00709-t002:** Summary of MCPyV viral load, LTAg DNA sequences and LTAg/VP1 transcripts detected in oral cavity tissues.

Case n°	Histological Diagnosis	MCPyV Viral Load (Copies/mL)	MCPyV LTAg		MCPyV Transcripts	
			LT1	LT3	LT	VP1
3	OSCC, tongue, G1	1.1 × 10^2^	+	+	+	+
7	OSCC, oral cavity, G3	1.17 × 10^2^	+	+	+	+
10	OSCC, oral cavity, G3	1.25 × 10^2^	+	+	+	+
14	keratosis with dysplasia	1.4 × 10^2^	+	+	+	+
15	keratosis without dysplasia	1.34 × 10^2^	+	+	+	+
17	keratosis without dysplasia	1 × 10^2^	+	+	+	+
22	OLP	3.2 × 10^2^	+	+	+	+

**Table 3 biomedicines-12-00709-t003:** Analysis of NCCR, VP1 and LTAg sequences in MCPyV positive samples.

Case n°	MCPyV NCCR Sequencing	MCPyV VP1 Sequencing	MCPyV LTAg Sequencing
3	canonical	canonical	full-length
7	canonical	4179 C to A transversion	full-length
10	canonical	canonical	full-length
14	canonical	canonical	full-length
15	canonical	canonical	full-length
17	canonical	4204 T to C transition	full-length
22	canonical	canonical	full-length

## Data Availability

Data is contained within the article.

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
