# Peer review of "Merkel Cell Polyomavirus in the Context of Oral Squamous Cell Carcinoma and Oral Potentially Malignant Disorders"

_biomedicines, 2024, doi:10.3390/biomedicines12040709_

Round 1

Reviewer 1 Report

Comments and Suggestions for Authors

The manuscript investigates the presence and molecular characteristics of Merkel Cell Polyomavirus (MCPyV) in oral squamous cell carcinoma (OSCC) and oral potentially malignant disorders (OPMDs). Through comprehensive analyses, including MCPyV DNA detection, sequence analysis, and miRNA expression profiling, the study sheds light on MCPyV's potential role in oral cavity lesions. The findings suggest evidence of MCPyV replication in both OSCC and OPMD specimens, indicating the oral cavity as a site of replicative MCPyV infection.

Here are some suggestions for improvement:

Minor comments: The introduction covers a broad range of topics related to oral cancer, viral oncogenesis, and the characteristics of MCPyV. Consider breaking down the information into more focused paragraphs to improve readability and clarity. Ensure a clear transition between different sections of the introduction, such as risk factors for oral cancer, the role of viruses in oncogenesis, and specific details about MCPyV. Additionally:

·        Clarify the significance of MCPyV in the context of Merkel Cell Carcinoma (MCC) before discussing its potential involvement in oral cancers.

·        Some sentences contain redundant phrases or unnecessary details. Streamline the text to convey information more concisely. Focus on key findings and implications related to MCPyV in OSCC and oral potentially malignant disorders (OPMDs).

·        Ensure proper citation of sources for statements or claims made in the introduction, especially when discussing previous research findings related to MCPyV and oral cancers.

·        Consider adding a brief paragraph at the end of the introduction to outline the objectives or hypotheses of the current study and how it contributes to the existing knowledge on MCPyV and oral cancers.

Major comments: The Results section provides a clear presentation of the findings, including the detection of MCPyV DNA, analysis of viral sequences, and examination of miRNA expression. However, a few in-depth mechanistic studies will further enrich this research manuscript as listed below: 

·        MCPyV Integration Analysis: Investigate MCPyV integration sites within the host genome using techniques such as whole-genome sequencing or fluorescence in situ hybridization (FISH). Determine the impact of MCPyV integration on host gene expression and cellular pathways involved in oral carcinogenesis.

·        Functional Characterization of MCPyV LT and sT Antigens: Conduct in vitro studies to elucidate the specific roles of MCPyV LT and sT antigens in oral cancer cell lines. Explore the molecular mechanisms underlying MCPyV-mediated oncogenesis, including interactions with host cell signaling pathways and regulators of cell proliferation and apoptosis.

·        miRNA Profiling and Functional Studies: What is the rational of checking the expression of miR-375 ambiguous. It would be better to perform a comprehensive miRNA profiling in MCPyV-positive and negative oral cavity tissues to identify dysregulated miRNAs associated with viral infection.

Overall, with these revisions, the manuscript will be strengthened in terms of clarity, focus, and completeness, leading to a more impactful presentation of the study findings.

Author Response

Reviewer 1

The manuscript investigates the presence and molecular characteristics of Merkel Cell Polyomavirus (MCPyV) in oral squamous cell carcinoma (OSCC) and oral potentially malignant disorders (OPMDs). Through comprehensive analyses, including MCPyV DNA detection, sequence analysis, and miRNA expression profiling, the study sheds light on MCPyV's potential role in oral cavity lesions. The findings suggest evidence of MCPyV replication in both OSCC and OPMD specimens, indicating the oral cavity as a site of replicative MCPyV infection.

Here are some suggestions for improvement:

Minor comments: The introduction covers a broad range of topics related to oral cancer, viral oncogenesis, and the characteristics of MCPyV. Consider breaking down the information into more focused paragraphs to improve readability and clarity. Ensure a clear transition between different sections of the introduction, such as risk factors for oral cancer, the role of viruses in oncogenesis, and specific details about MCPyV. Additionally:

  • Clarify the significance of MCPyV in the context of Merkel Cell Carcinoma (MCC) before discussing its potential involvement in oral cancers.
  • Some sentences contain redundant phrases or unnecessary details. Streamline the text to convey information more concisely.Focus on key findings and implications related to MCPyV in OSCC and oral potentially malignant disorders (OPMDs).
  • Ensure proper citation of sources for statements or claims made in the introduction, especially when discussing previous research findings related to MCPyV and oral cancers.
  • Consider adding a brief paragraph at the end of the introduction to outline the objectives or hypotheses of the current study and how it contributes to the existing knowledge on MCPyV and oral cancers.

We agree with the reviewer and have changed the end of the introduction as follows:

“Because the incidence and possible role of MCPyV in oral cancer remains elusive, the purpose of our study was to investigated the presence of the viral DNA and RNA and to examine integration and truncation of LT, two hallmarks of MCPyV-positive MCCs, in OSCC and OPMD. Although MCPyV DNA was present and viral genes were expressed in some samples, the absence of integrated MCPyV and truncated LT may jeopardize a role in oral cancer or suggest a different modus operandi in MCPyV-induced oral cancer”.

Major comments: The Results section provides a clear presentation of the findings, including the detection of MCPyV DNA, analysis of viral sequences, and examination of miRNA expression. However, a few in-depth mechanistic studies will further enrich this research manuscript as listed below: 

  • MCPyV Integration Analysis: Investigate MCPyV integration sites within the host genome using techniques such as whole-genome sequencing or fluorescence in situ hybridization (FISH). Determine the impact of MCPyV integration on host gene expression and cellular pathways involved in oral carcinogenesis.

We agree with the reviewer that integration of MCPyV may disrupt a cellular gene and/or the MCPyV promoter may have an impact on the expression of adjacent cellular genes but it is difficult to determine the impact of integration on host gene expression and cellular pathways because the viral oncoproteins may be responsible for altered host gene expression and signaling pathways rather than the host gene disrupted by viral integration. If one wants to determine the impact of solely MCPyV integration on these processes, integration of a MCPyV variant with inactivated LT and sT antigens should be performed.

We absolutely agree with the reviewer that determining the site of integration and the number of viral genome copies would give important information on the possible role of MCPyV in oral cancer. However, in none of our MCPyV-positive samples integrated viral genome was detected. Therefore, performing FISH and whole genome sequencing is not relevant and discussing the impact of integration is highly speculative.

  • Functional Characterization of MCPyV LT and sT Antigens: Conduct in vitro studies to elucidate the specific roles of MCPyV LT and sT antigens in oral cancer cell lines. Explore the molecular mechanisms underlying MCPyV-mediated oncogenesis, including interactions with host cell signaling pathways and regulators of cell proliferation and apoptosis.

The purpose of this study was to investigate the presence of MCPyV in OSCC and OMPD samples and the expression of viral genes and the state (integrated or episomal) of the virus. This is a first step in exploring a possible role of MCPyV in oral cancer. The studies suggested by the reviewer are necessary to establish a possible role of this virus in oral cancer and to elucidate the mechanisms by which MCPyV may induce this type of cancer. However, the studies suggested by the reviewer are beyond the scope of the current study. In vitro experiments are in progress on primary oral cells in order to better understand MCPyV molecular biology and oncogenic potential in oral cavity, and the obtained results will be object of further publications.

  • miRNA Profiling and Functional Studies: What is the rational of checking the expression of miR-375 ambiguous. It would be better to perform a comprehensive miRNA profiling in MCPyV-positive and negative oral cavity tissues to identify dysregulated miRNAs associated with viral infection.

miR-375 profiling was done because this miR has been studied in MCC and was shown to be highly expressed and to generate a pro-tumorigenic microenvironment by inducing fibroblast polarization (Fan et al. Oncogene. 2021; 40:980-996. Doi: 10.1038/s41388-020-01576-6). Specifically, miR-375 levels were significantly higher in virus positive than the virus negative MCC tumors, suggesting its potential use as a surrogate marker for tumor burden in MCC. Moreover, deregulated expression of miR-375 in oral cancer has been reported (for a recent review see Dioguardi et al. Noncoding RNA. 2023; 9:54. Doi: 10.3390/ncrna9050054). We agree with the reviewer that comparing the miR profile in our virus-positive and virus-negative samples would be more informative but this was not the scope of our study.

Overall, with these revisions, the manuscript will be strengthened in terms of clarity, focus, and completeness, leading to a more impactful presentation of the study findings.

We agree with the reviewer that all these studies are required to elucidate a possible role for MCPyV in oral cancer. However, the purpose of this study was to investigate whether MCPyV can be detected in these types of cancer. Rather than just performing PCR to detect viral DNA, we also monitored viral expression (early and late genes and viral miR) and the state of the virus (integrated or episomal).

Reviewer 2 Report

Comments and Suggestions for Authors

This study analyzed Merkel Cell Polyomavirus in oral squamous cell carcinoma and other lesions, and found around 30% of the cases were positive. Usually, this virus is associated to Merkel Cell Carcinoma of the skin, but some cases have been described in oral region. The text is well written, it is easy to read and understand.

Comments:

(1) In the abstract "MCPyV DNA was found in 3 OSCC (n=11) and 4 OPMD (n=12) samples. Does it mean that the sample size of the study was 11 OSCC and 12 OPMD?

(2) Lines 52-61. The structure of the virus, including the MCPyV genome could be depicted in a figure.

You may refer to:

https://www.tumorvirology.pitt.edu/our-viruses/merkel-cell-polyomavirus

Houben R, Celikdemir B, Kervarrec T, Schrama D. Merkel Cell Polyomavirus: Infection, Genome, Transcripts and Its Role in Development of Merkel Cell Carcinoma. Cancers (Basel). 2023;15(2):444. Published 2023 Jan 10. doi:10.3390/cancers15020444

(3) Regarding MYCL, please add the complete name as MYCL Proto-Oncogene, BHLH Transcription Factor. If the pathogenic pathway could be added in figure, this would help to understand better the mechanisms.

(4) Regarding MCPyV. You may emphasizen that Merkel-cell carcinoma is a rare and aggressive skin cancer occurring in about three people per million members of the population. It is also known as cutaneous APUDoma, primary neuroendocrine carcinoma of the skin, primary small cell carcinoma of the skin, and trabecular carcinoma of the skin.

(5) In the introduction, you may add that MCC can also be found in oral region.

Islam MN, Chehal H, Smith MH, Islam S, Bhattacharyya I. Merkel Cell Carcinoma of the Buccal Mucosa and Lower Lip. Head Neck Pathol. 2018;12(2):279-285. doi:10.1007/s12105-017-0859-1   Yom SS, Rosenthal DI, El-Naggar AK, Kies MS,

Hessel AC. Merkel cell carcinoma of the tongue and head and neck oral mucosal sites. Oral Surg Oral Med Oral Pathol Oral Radiol Endod. 2006;101(6):761-768. doi:10.1016/j.tripleo.2005.10.068   Sheldon JD, Lott Limbach AA.

Merkel Cell Carcinoma of the Maxillary Sinus: An Unusual Presentation of a Common Tumor. Head Neck Pathol. 2021;15(2):691-697. doi:10.1007/s12105-020-01219-y

(6) Line 135. It looks like IRB was not necessary. Please add the Helsinki Declaration statement.  

(7) Regarding the histological diagnosis (lines 140-151). It may be useful to describe  the histological characteristics of each type of lesion so the authors do not need to refer to the bluebook  

(8) In materials and methods. Please add the catalog number of the reagents (where appropriate).  

(9) Please add the primers (lines 159-160).  

(10) Is there any of the agarose gel images available (line 170)?  

(11) Figure 2A shows the viral loads. What would be expected from a Merkel cell carcinoma?  

(12) In 2b, what type of lesion are included in the pre-malignant and malignant grous?  

(13) Should the viral loads have mean and standard deviation?

(14) As I understand from table 2. There were no differences between the different histological subtype of lesions? Only oral lichen planus had more viral load?

(15) Is skin OSCC, is the MCPyV also positive in a percentage of cases? Is there data?

(16) It would be nice if immunohistochemistry could be performed in this series. To confirm the expression i epithelial cells. Recombinant Anti-Polyoma virus, Large T antigen antibody [CM2B4] (ab307450)

(17) Have you analyzed normal control negative oral mucosa? Would you expect normal mucosa be infected with this virus in around 30% of the cases?

Author Response

Reviewer 2

This study analyzed Merkel Cell Polyomavirus in oral squamous cell carcinoma and other lesions, and found around 30% of the cases were positive. Usually, this virus is associated to Merkel Cell Carcinoma of the skin, but some cases have been described in oral region. The text is well written, it is easy to read and understand.

Comments:

(1) In the abstract "MCPyV DNA was found in 3 OSCC (n=11) and 4 OPMD (n=12) samples. Does it mean that the sample size of the study was 11 OSCC and 12 OPMD?

Indeed. We have changed the text as follows:

MCPyV DNA was found in 3 out of 11 OSCC and 4 out of 12 OPMD samples, with a viral mean value of 1.49x102 copies/ml.

(2) Lines 52-61. The structure of the virus, including the MCPyV genome could be depicted in a figure.

The structure and genome of MCPyV has been presented in several review articles. As this is not a review article but an original article and therefore it is not the custom to present such a figure. Moreover, copyright has to be acquired from the journal that published the figure. It is not sufficient to refer to the authors and their article.

You may refer to:

https://www.tumorvirology.pitt.edu/our-viruses/merkel-cell-polyomavirus

Houben R, Celikdemir B, Kervarrec T, Schrama D. Merkel Cell Polyomavirus: Infection, Genome, Transcripts and Its Role in Development of Merkel Cell Carcinoma. Cancers (Basel). 2023;15(2):444. Published 2023 Jan 10. doi:10.3390/cancers15020444

(3) Regarding MYCL, please add the complete name as MYCL Proto-Oncogene, BHLH Transcription Factor. If the pathogenic pathway could be added in figure, this would help to understand better the mechanisms.

We added the complete name in the text. As previously said for MCPyV genome, since this is not a review article but an original article, it is not the custom to present such a figure.

(4) Regarding MCPyV. You may emphasizen that Merkel-cell carcinoma is a rare and aggressive skin cancer occurring in about three people per million members of the population. It is also known as cutaneous APUDoma, primary neuroendocrine carcinoma of the skin, primary small cell carcinoma of the skin, and trabecular carcinoma of the skin.

(5) In the introduction, you may add that MCC can also be found in oral region.

Islam MN, Chehal H, Smith MH, Islam S, Bhattacharyya I. Merkel Cell Carcinoma of the Buccal Mucosa and Lower Lip. Head Neck Pathol. 2018;12(2):279-285. doi:10.1007/s12105-017-0859-1   Yom SS, Rosenthal DI, El-Naggar AK, Kies MS,

Hessel AC. Merkel cell carcinoma of the tongue and head and neck oral mucosal sites. Oral Surg Oral Med Oral Pathol Oral Radiol Endod. 2006;101(6):761-768. doi:10.1016/j.tripleo.2005.10.068   Sheldon JD, Lott Limbach AA.

Merkel Cell Carcinoma of the Maxillary Sinus: An Unusual Presentation of a Common Tumor. Head Neck Pathol. 2021;15(2):691-697. doi:10.1007/s12105-020-01219-y

We thank the reviewer for these references and have included them in the revised manuscript.

(6) Line 135. It looks like IRB was not necessary. Please add the Helsinki Declaration statement.  

As we wrote in the paragraph about clinical specimens, “This study was conducted retrospectively from data obtained for clinical purposes”, therefore in this case the Ethic Committee Sapienza University of Rome, Policlinico Umberto I determined that this type of study did not need ethical approval and Helsinki Declaration statement.  

(7) Regarding the histological diagnosis (lines 140-151). It may be useful to describe the histological characteristics of each type of lesion so the authors do not need to refer to the bluebook  

We thank the reviewer for this suggestion and added a more detailed description of the histological characteristics of oral lesions.

(8) In materials and methods. Please add the catalog number of the reagents (where appropriate).  

In materials and methods, as usual in research articles, we reported the reagents’ manufacturer and its country of origin.

(9) Please add the primers (lines 159-160).  

These primers have been used in several other studies and their sequence can be found in the references. However, as you kindly suggest, we added the forward and reverse primer sequences in the text.  

(10) Is there any of the agarose gel images available (line 170)?  

Yes, we added the available gel images as supplementary materials.

(11) Figure 2A shows the viral loads. What would be expected from a Merkel cell carcinoma?  

For MCC, viral load is often expressed as genome copies/cell and usually >1/cell (e.g. Feng et al. Science. 2008; 319:1096-1100. Doi: 10.1126/science.1152586; Martel-Jantin et al. Virology. 2012; 426:134-142. Doi: 10.1016/j.virol.2012.01.018). In the current study we expressed the viral genome copies as copies/ml reporting about 1x102 copie/ml as mean viral load in our samples. Calculating approximately the number of copies/cell we can assume to have <<1 copies/cell, about 0.2 copies/cell as mean viral load in our MCPyV positive oral biopsies.

 (12) In 2b, what type of lesion are included in the pre-malignant and malignant groups?  

We have changed the figure 2b, renaming the two groups as “OSCC” and “OPMD” to make the results clearer.

(13) Should the viral loads have mean and standard deviation?

We added standard deviation to viral loads in the results.

(14) As I understand from table 2. There were no differences between the different histological subtype of lesions? Only oral lichen planus had more viral load?

Yes, we found similar viral load among the different histological subtype of lesions. Only in oral lichen planus was observed a higher viral load. However, due to the small sample size and since only one out of the 4 analyzed oral lichen planus specimen was found positive for MCPyV, we cannot make particular assumptions about viral replication in this type of oral lesion.

(15) Is skin OSCC, is the MCPyV also positive in a percentage of cases? Is there data?

Yes. MCPyV has been detected in about 24,9% squamous cell carcinomas (SCCs) and in 4% combined basal cell carcinoma (BCC) with SCC.

(16) It would be nice if immunohistochemistry could be performed in this series. To confirm the expression in epithelial cells. Recombinant Anti-Polyoma virus, Large T antigen antibody [CM2B4] (ab307450)

We completely agree with the reviewer. Unfortunately, due to the small size of the samples there was not enough material to do IHC in addition to the DNA, RNA and integration analysis.

(17) Have you analyzed normal control negative oral mucosa? Would you expect normal mucosa be infected with this virus in around 30% of the cases?

No, we did not analyze samples from normal control negative oral mucosa. However, basing on the reviewer suggestion, it could be interesting for further studies to in parallel investigate MCPyV prevalence in oral lesion and normal control oral mucosa.

Reviewer 3 Report

Comments and Suggestions for Authors

The article “Merkel Cell Polyomavirus in the context of Oral Squamous  Carcinoma and Oral Potentially Malignant Disorders” is interesting, I have to make some comments with the intention of improving it.

The strength of the study is the novelty and new contributions in relation to an uncommon topic, but the presentation of the methodology and results must be improved.

 - The authors first classify OPMD lesions into keratosis with/withouth dysplasia, proliferative verrucous leukoplakia and lichen planus, however in the results, in figure 2 (a) you show OLP, and lichenoid (withouth dysplasia and with low grade dysplasia) or pre-malignant (in figure b).

 The authors must clarify the discrepancy in the nomenclature to properly interpret the results, especially miR-375, and which you include in OPMD and in pre-malignant lesions (clinical or histological).

- How authors did the lichen/lichenoid or keratosis classification, and the dysplasia criteria? You must reflect it in the material and method.

Thank you

Author Response

Reviewer 3

The article “Merkel Cell Polyomavirus in the context of Oral Squamous Carcinoma and Oral Potentially Malignant Disorders” is interesting, I have to make some comments with the intention of improving it.

The strength of the study is the novelty and new contributions in relation to an uncommon topic, but the presentation of the methodology and results must be improved.

 - The authors first classify OPMD lesions into keratosis with/withouth dysplasia, proliferative verrucous leukoplakia and lichen planus, however in the results, in figure 2 (a) you show OLP, and lichenoid (withouth dysplasia and with low grade dysplasia) or pre-malignant (in figure b).

 The authors must clarify the discrepancy in the nomenclature to properly interpret the results, especially miR-375, and which you include in OPMD and in pre-malignant lesions (clinical or histological).

As the reviewer kindly suggest, in order to properly interpret the results, we conform the nomenclature in the results with that reported in the previous paragraphs. Therefore, we changed “lichen planus” in “oral lichen planus”, adding also the abbreviation “OLP” in the text and we have changed the figure 2b, renaming the two groups as “OSCC” and “OPMD” without making distinction in malignant and pre-malignant.

- How authors did the lichen/lichenoid or keratosis classification, and the dysplasia criteria? You must reflect it in the material and method.

We thank the reviewer for this observation and added a more detailed description of the histological diagnosis in the material and methods.

Thank you

Round 2

Reviewer 1 Report

Comments and Suggestions for Authors

Thank you for addressing all the raised concerns.